# Taxonomic Composition and Diversity of the Gut Microbiota in Relation to Habitual Dietary Intake in Korean Adults

**DOI:** 10.3390/nu13020366

**Published:** 2021-01-26

**Authors:** Hwayoung Noh, Hwan-Hee Jang, Gichang Kim, Semi Zouiouich, Su-Yeon Cho, Hyeon-Jeong Kim, Jeongseon Kim, Jeong-Sook Choe, Marc J. Gunter, Pietro Ferrari, Augustin Scalbert, Heinz Freisling

**Affiliations:** 1Nutrition and Metabolism Branch, International Agency for Research on Cancer (IARC-WHO), 69008 Lyon, France; hwayoung.noh@inserm.fr (H.N.); ZouiouichS@students.iarc.fr (S.Z.); GunterM@iarc.fr (M.J.G.); FerrariP@iarc.fr (P.F.); ScalbertA@iarc.fr (A.S.); 2Department of Cancer Prevention and Environment, INSERM UA8, Léon Bérard Cancer Center, 69003 Lyon, France; 3National Institute of Agricultural Sciences, Rural Development Administration (NAS-RDA), Wanju, Jeollabuk-do 55365, Korea; rapture19@korea.kr (H.-H.J.); recall@korea.kr (G.K.); 0727jsy@naver.com (S.-Y.C.); mamimumemo14@hanmail.net (H.-J.K.); swany@korea.kr (J.-S.C.); 4Department of Cancer Biomedical Science, Graduate School of Cancer Science and Policy, National Cancer Center, Goyang 10408, Korea; jskim@ncc.re.kr

**Keywords:** human gut microbiota, enterotypes, habitual diet, dietary pattern, Korean adults

## Abstract

We investigated associations of habitual dietary intake with the taxonomic composition and diversity of the human gut microbiota in 222 Koreans aged 18–58 years in a cross-sectional study. Gut microbiota data were obtained by 16S rRNA gene sequencing on DNA extracted from fecal samples. The habitual diet for the previous year was assessed by a food frequency questionnaire. After multivariable adjustment, intake of several food groups including vegetables, fermented legumes, legumes, dairy products, processed meat, and non-alcoholic beverages were associated with major phyla of the gut microbiota. A dietary pattern related to higher α-diversity (HiαDP) derived by reduced rank regression was characterized by higher intakes of fermented legumes, vegetables, seaweeds, and nuts/seeds and lower intakes of non-alcoholic beverages. The HiαDP was positively associated with several genera of *Firmicutes* such as *Lactobacillus*, *Ruminococcus*, and *Eubacterium* (all *p* < 0.05). Among enterotypes identified by principal coordinate analysis based on the β-diversity, the *Ruminococcus* enterotype had higher HiαDP scores and was strongly positively associated with intakes of vegetables, seaweeds, and nuts/seeds, compared to the two other enterotypes. We conclude that a plant- and fermented food-based diet was positively associated with some genera of *Firmicutes* (e.g., *Lactobacillus*, *Ruminococcus*, and *Eubacterium*) reflecting better gut microbial health.

## 1. Introduction

The human gut microbiota is a complex community consisting of 10^13^–10^14^ microorganisms, dominated by bacteria, which inhabit the human gastrointestinal tract [1]. The volume of the collective microbial genome is over 100 times larger than the human genome [1,2]. In a symbiotic relationship with the host, the gut microbiota contributes to numerous physiological processes, such as modulating the intestinal gut barrier [3], regulating energy metabolism [4,5], protection against pathogens [6], and regulating the immune system [7].

Host dietary intake is one of the main factors that can modulate the taxonomic composition (classified groups of closely related microbiota) and diversity (distribution of microbiota within or between communities) of the gut microbiota, which could, in turn, promote either beneficial or detrimental consequences on host health through alterations of the physiological functions of the gut microbiota [8,9,10]. Diets rich in animal-based foods such as “Western-style diet” can increase the levels of bile-tolerant bacteria including *Bacteroidetes* (e.g., *Bacteroides* and *Alistipes*), *Proteobacteria* (*Bilophila*) and decrease levels of fiber-degrading bacteria such as *Firmicutes* (e.g., *Eubacterium* and *Ruminococccus*) [11,12]. In contrast, plant-based diets such as the “Mediterranean diet” can promote fiber-degrading bacteria, mainly including genera of the *Firmicutes* phylum, together with increased overall diversity of the gut microbiota [13,14,15].

The majority of studies that have investigated associations between diet and the gut microbiota to date have focused on “Western-style” or Mediterranean diets and have been conducted mainly in European and American populations [15]. In contrast, little is known about the associations between dietary habits and the gut microbiota in the Korean population [16]. Traditional Korean diets are characterized by higher intakes of fermented vegetables, e.g., kimchi, and legumes, e.g., fermented soybean [17,18]. Fermented foods are known to contain large amounts of microorganisms, and their strains are phylogenetically similar to probiotic strains, which could affect the composition and diversity of the gut microbiota, thus affecting human health [19,20].

Recent data from the International Human Microbiome Consortium (IHMC) and the European Metagenomics of the Human Intestinal Tract (MetaHIT) consortium have indicated that the human gut microbiota could be classified into distinct “enterotypes” [8]. Each of the three identified enterotypes was distinguished by different microbial composition at the genus level, with prominent variation in *Bacteroides*, *Prevotella*, and *Ruminococcus*. So far, only two studies have examined associations between these enterotypes and habitual diets in American [21] and in Korean [16] adults.

In a collaborative study between the National Institute of Agricultural Sciences of Korea and the International Agency for Research on Cancer (NAS—IARC), we investigated associations of long-term intake of both foods and nutrients with the taxonomic composition and diversity of the gut microbiota in Korean adults aged 18–60 years. We also aimed to identify dietary patterns associated with gut microbial within-sample (α-) diversity and to explore whether different enterotypes based on gut microbial between-sample (β-) diversity were associated with long-term dietary intake.

## 2. Materials and Methods

### 2.1. Study Design and Subjects

Within the NAS-IARC cross-sectional study, participants were residents aged 18–60 years in the local vicinity (within 20 km) of the NAS, the Republic of Korea, between March and October 2018. We excluded volunteers who, prior to recruitment (1) were underweight (body mass index (BMI) < 18.5 kg/m^2^) or obese (BMI ≥ 30 kg/m^2^), (2) reported any chronic disease such as inflammatory bowel disease, hypertension, diabetes, hyperlipidemia, or cancer, (3) had taken medication including antibiotics within the past 2 weeks, (4) had taken hormone replacement therapy or used oral contraceptives within the past 2 weeks, or (5) were pregnant or breastfed within the past 6 months. Volunteers who had taken any dietary supplements within the past 3 months were not excluded, but this information was collected using lifestyle questionnaires. The study participants were initially invited to an information meeting, approximately one week prior to the start of the study, where anthropometric data including height and weight were measured by trained research assistants, and exclusion criteria were ascertained. Those eligible for the study were provided with a lifestyle questionnaire (physical activity, alcohol intake, smoking, and socioeconomic status) and a food frequency questionnaire (FFQ) with instructions, and were asked to fill in and return on the study day. During the study day, on-site fecal samples were collected and FFQ and lifestyle data of participants were reviewed by trained research assistants following standardized protocols. Of a total of 229 eligible participants, seven participants failed to collect fecal samples, leading to a sample size of 222 healthy Korean adults (49% males) for this study. We confirmed that all subjects had not taken antibiotics within the past 3 months prior to recruitment.

All procedures and protocols of the study were approved by the Public Institutional Review Boards of the Ministry of Health and Welfare, Korea (Approval no: P01-201801-11-003), and were registered at the WHO International Clinical Trials Registry Platform (ICTRP) (http://apps.who.int/trialsearch/; Registration No.: KCT0002831). All study participants provided written informed consent.

### 2.2. Dietary Data Collection

Long-term dietary intake data from participants were collected with a semi-quantitative FFQ, which was developed and validated for the Korean diet by the Korea National Institute of Health (KNIH) [22]. The FFQ included 106 food/dish items, including nine Korean staple dishes (rice and noodles), 25 soups and stews, 54 side dishes, nine non-alcoholic beverages, and nine fruits. Subjects were asked to report the consumption frequency and average portion size of each item during the previous year. During the visit of the participants, trained research assistants reviewed the questionnaires with participants for completeness. The 106 food/dish items were classified into 25 food groups—potatoes, vegetables, fermented vegetables, seaweeds, legumes, fermented legumes, fruit/fruit juice, nuts/seeds, dairy, refined grains, multi/whole grains, other cereal products, meats, processed meats, fish/seashells, eggs, vegetable oils, other fats, sugar/confectionery, cakes/sweets, coffee, tea, non-alcoholic beverages, pizza/burgers, and salty snacks based on their recipe. In particular, vegetable and legume groups were divided into two sub-groups, non-fermented and fermented, to take fermentation into account, which could affect gut microbial composition and diversity. This classification of the food groups is shown in Appendix A. Intakes of macronutrients including protein, fat, carbohydrates (CHO), and dietary fiber were also estimated based on the FFQ data. Protein and fat intake were classified as either plant-based or animal-based separately. Additionally, saturated fatty acids (SFA), monounsaturated fatty acids (MUFA), and polyunsaturated fatty acids (PUFA) were estimated separately. The intakes of food groups and macronutrients were calculated as gram per day (g/day) based on the consumption frequency and average portion size according to a food composition database established for the FFQ [22]. Alcohol intake of the previous year was collected with a lifestyle questionnaire and converted into g/day.

### 2.3. Fecal Sample Collection

The fecal specimens were collected on-site on the study day at the NAS. We provided a collection tube (SARSTEDT AG & Co., Nümbrecht, Germany) for the fecal sample to each participant. Following the collection, the samples were immediately delivered to the laboratory for processing. Each fecal specimen was mixed manually using a spatula, and approximately 1–2 g of feces for each participant was aliquoted, representing a full scoop of feces, into stool nucleic acid collection tubes (Norgen Biotek Co., Thorold, ON, Canada). Samples were then frozen and stored at 4 °C until further processing (average time between sample collection and storage: approx. 12 min).

### 2.4. 16S rRNA Gene Sequencing and Taxonomic Assignment

All procedures, from extracting bacterial DNA from the collected fecal samples to generating the gut microbial composition and diversity data, were performed by a biotechnology company (Macrogen Inc., Seoul, Korea) in Seoul, the Republic of Korea. On a weekly basis, the fecal samples collected for one week were transferred to Macrogen Inc., and bacterial DNA from each sample was extracted using PowerSoil^®^ DNA Isolation Kit (Cat. No. 12888, MO BIO) and stored at −80 °C until all samples were collected for further analysis. DNA quantity and quality were measured by PicoGreen and Nanodrop (ThermoFisher Sci. Inc. Waltham, MA, USA). The 16S rRNA amplicons covering variable regions V3-V4 were generated using the primers (forward: 5′-TCGTCGGCAGCGTCAGA TGTGTATAAGAGACAGCCTACGGGNGGCWGCAG-3′ and reverse: 5′-GTCTCGTGG GCTCGGAGATGTGTATAAGAGACAGGACTACHVGGGTATCTAATCC-3′). The final products were normalized and pooled using PicoGreen, the size of libraries was verified using TapeStation DNA screen-tape D1000 (Agilent Tech., Santa Clara, CA, USA), and the amplicons were sequenced using the MiSeq™ platform (Illumina, San Diego, CA, USA). In order to achieve a high quality of data on Illumina sequencing platforms, the pre-processing of the sequencing data was conducted as follows: (1) adapters were trimmed using the SeqPurge [23], (2) the length of short reads was adjusted by overlapping and merging paired-end reads using FLASH (1.2.11) [24], (3) sequencing errors were removed by identifying and removing low-quality reads, ambiguous reads, and chimeric reads using CD-HIT-OTU [25], and (4) using the QIIME 1.9.0 pipeline [26], the sequences were clustered into operational taxonomic units (OTUs) from phylum to species levels with 97% identity using CD-HIT-OTU. The 16S rRNA gene sequencing data supporting the conclusion of this study are available in NCI Sequence Read Archive (SRA) with study accession number: PRJNA644479.

In total, 5.7 million sequence reads from 222 subjects were obtained with an average of 25,852 reads per subject. For standardization, the sequence reads of each subject were rarefied to the minimum of sequence reads (5887 reads) within total subjects [27], and then were clustered into OTUs, and subsequently assigned taxonomy at different levels. The gut microbial taxonomic composition and diversity data for statistical analyses included individual-level information on (1) relative abundance (proportion (%) of OTU) at different bacterial taxonomic levels from phylum to genus levels; (2) within-sample (α-) diversity to understand the number (richness) and/or distribution (evenness) of species within a single sample by estimating three different α-diversity indices [28]—(i) Chao1 index, an abundance-based index of species richness [29,30], (ii) Shannon index, an index of both species richness and evenness [31], and (iii) Faith’s phylogenetic diversity (Faith PD), an index of phylogenetic standardized species richness [32,33]; and (3) between-sample (β-) diversity to understand differences of gut microbial composition in one subject compared to another by measuring three different β-diversity distance matrixes [28]—(i) Bray-Curtis, a distance matrix considering the relative abundances of species [34] and (ii) weighted and (iii) unweighted UniFrac, phylogenetic distance matrixes considering the presence/absence of species with and without weighing the relative abundances [35], respectively.

### 2.5. Statistical Analysis

Dietary intake data were log-transformed to render the distributions symmetrical and to approximate normality and were adjusted for total energy intake using the residual method. The taxonomic composition data were centered log-ratio (clr) transformed after imputing zeros in the dataset based on a Bayesian-multiplicative replacement [36]. The α-diversity indices were also log-transformed. The differences of relative abundance (% OTU) of the four major phyla and the *Firmicutes-to-Bacteroidetes* (*F/B*) ratio, which are the two major phyla in human gut microbiota and are known to be modulated by diet [9,11], by basic characteristics and lifestyle factors (sex; age group: <40 years vs. ≥40 years; BMI group: <23 kg/m^2^ vs. ≥23 kg/m^2^; dietary supplement intake within 3 months prior to the enrolment: yes vs. no; regular physical activity: yes vs. no; smoking status: ever vs. never, education: < university graduation vs. ≥ university graduation; household income: <4000 USD/month vs. ≥4000 USD/month) were examined by Wilcoxon-Mann-Whitney tests. Associations of within-sample (α-) diversity and between-sample (β-) diversity of the gut microbiota with characteristics of the study populations were examined by general linear models (GLMs) and permutational multivariate analysis of variance (PERMANOVA), respectively.

In order to examine the gut microbial composition in relation to dietary intake, partial Spearman’s correlation coefficients of the clr transformed relative abundance of the four major phyla, and the clr transformed *F/B* ratio of human gut microbiota with intakes of food groups and macronutrients were estimated and correlation values displayed using heatmaps. Adjustment for sex, age, BMI, dietary supplement intake, physical activity, smoking status, and sample batch was performed.

Partial Spearman’s correlation coefficients of the within-sample (α-) diversity indices with the intakes of food groups and macronutrients were estimated after adjustment for sex, age, BMI, dietary supplement intake, physical activity, smoking status, and sample batch. To identify dietary patterns associated with high α-diversity (HiαDPs), reduced rank regression (RRR) models were used to derive linear combinations of 25 food groups (predictor variables) maximizing the explained variability of gut microbiota within-sample diversity (each α-diversity index (Chao1, Shannon, and Faith PD) as a response variable in each RRR model) [37]. We then examined partial Spearman’s correlation coefficients between the α-diversity dietary pattern (HiαDP score) and the clr transformed relative abundance of major phyla, including *F/B* ratio and genera within the major phyla of the human gut microbiota, with sex, age, BMI, dietary supplement intake, physical activity, smoking status, and sample batch as covariates and with multiple comparison corrections using false discovery rate (FDR). In addition, partial Spearman’s correlations between main contributing food groups (factor loading > 0.3) of the identified HiαDPs and the clr transformed relative abundance of genera within the two major phyla were examined. Adjustments were made for sex, age, BMI, dietary supplement intake, physical activity, smoking status, and sample batch, and multiple comparison corrections were made using FDR.

Enterotypes of gut microbiota in healthy Korean adults were explored by a modified method to determine enterotype discovery in the previous study [8] with a combination of principal coordinate analysis (PCoA) based on between-sample (β-) diversity indices (unweighted and weighted UniFrac and Bray-Curtis), and then k-means cluster analysis based on the PCoA scores of the first two principal coordinates (PCos). The optimal number of clusters was determined by visual inspection of clusters derived by three different methods—Elbow [38], Silhouette [39], and Gap statistic [40] methods (Appendix A) and by a priori knowledge [8]. The differences in general characteristics and lifestyle factors by enterotypes were examined by GLMs for continuous variables and chi-square test for categorical variables, and the differences in dietary intake, the HiαDP score and intakes of food groups and macronutrients, by enterotypes were examined by GLMs with sex, age, BMI, dietary supplement intake, physical activity, smoking status, and sample batch as covariates.

All analyses were performed using the R statistical software (version 3.6.1, R Development Core Team, 2019) for zero-imputation based on a Bayesian-multiplicative replacement, clr transformation, PCoA and k-means cluster analyses (using cmultRepl, clr, cmdscale, kmeans, and fviz_nbcluster functions) and generating heatmaps and boxplots, and SAS (version. 9.4, The SAS Institute, Cary, NC, USA) for the rest of the analyses.

## 3. Results

A total of 222 Korean adults (49% males) aged 18~58 years were included in this study. The main characteristics of the study population are shown in Table 1. The mean BMI of the study population was 22.9 kg/m^2^ (5–95 percentiles: 19.1–28.5 kg/m^2^), and was slightly higher in males than in females.

The dominant phyla in the study population were *Bacteroidetes*, *Firmicutes*, *Proteobacteria*, and *Actinobacteria*, of which medians of the relative abundance (% OTUs) were 54.2%, 37.6%, 3.8%, and 0.4%, respectively, and encompassed a total of 96.0% of the overall microbiota (Appendix A). The relative abundance of these four major phyla and the *F/B* ratio by general characteristics and lifestyle factors of the study population are shown in Appendix A. The *F/B* ratio was significantly higher in females than males, with a higher abundance of *Firmicutes* (39.8% vs. 32.8%, *p*-value = 0.009) and in never-smokers compared to ever-smokers, with a lower abundance of *Bacteroidetes* (53.2% vs. 57.4%, *p*-value = 0.017). Sex, age, and dietary supplement intake were all significantly associated with the abundance of *Proteobacteria*. Sex, age, BMI, physical activity, and smoking status were also significantly associated with within-sample (α-) and/or between-sample (β-) diversity of the gut microbiota, even though there were differences in the associations depending on indices (Appendix A). 

### 3.1. Association of Dietary Intake with Gut Microbial Composition

Heatmaps on partial Spearman correlations of the clr transformed relative abundance of four major phyla and the clr transformed *F/B* ratio of the gut microbiota with intakes of 25 food groups and macronutrients are shown in Figure 1. At the food group level (Figure 1A), intakes of vegetables (*r* = 0.19), fermented legumes (*r* = 0.16), vegetable oils (*r* = 0.15), potatoes (*r* = 0.14), and nuts/seeds (*r* = 0.14) were positively correlated with the *F/B* ratio, even though these were not significantly correlated with the relative abundance of *Firmicutes*. In contrast, intakes of non-alcoholic beverages including mainly carbonated and sweet beverages (*r* = −0.17) and other cereal products such as noodles (*r* = −0.12) were inversely correlated with the *F/B* ratio, but only intake of noodle products was inversely correlated with the relative abundance of *Firmicutes* (*r* = −0.16). Intake of processed meats was positively correlated with the relative abundance of *Bacteroidetes* (*r* = 0.15). Intakes of legumes (*r* = 0.14) and dairy products (*r* = 0.13) were correlated with the higher relative abundance of *Actinobacteria*. At the nutrient level (Figure 1B), intake of PUFA was positively correlated with the relative abundance of *Firmicutes* and the *F/B* ratio (both *r* = 0.19). Also, intakes of dietary fiber (*r* = 0.17) and plant protein (*r* = 0.15) were positively correlated with the *F/B* ratio. Intake of plant fat (*r* = 0.13) was positively correlated with the relative abundance of *Actinobacteria*.

### 3.2. Association of Dietary Intake with the within-Sample Diversity of Gut Microbiota

The correlations of intakes of food groups and macronutrients with different α-diversity indices are shown in Appendix A. Intakes of fermented legumes (*r* = 0.20), vegetables (*r* = 0.20), potatoes (*r* = 0.20), and seaweeds (*r* = 0.15) among food groups, and dietary fiber (*r* = 0.19) among nutrients, were all positively correlated with the Shannon index, but not with other α-diversity indices. Based on the Shannon index, we then identified a dietary pattern that best explained α-diversity of the gut microbiota (Figure 2 and Appendix A). This high α-diversity dietary pattern (HiαDP) was characterized by greater intakes of fermented legumes, potatoes, vegetables, seaweeds, nuts/seeds, and tea and lower intakes of non-alcoholic beverages (e.g., carbonated and sweet beverages). The HiαDP was inversely correlated with *Bacteroidetes* (*r* = −0.17, *p* < 0.001), but positively correlated with the *F/B* ratio (*r* = 0.24, *p* < 0.001). At the genus level, the HiαDP was inversely correlated with *Coenonia* (*r* = −0.27), *Prevotella* (*r* = −0.22)*, and*
*Tannerella*
*(r* = −0.22) within the *Bacteroidetes* phylum, but positively correlated with *Lactobacillus* (*r* = 0.23), *Ruminococcus* (*r* = 0.21), and *Eubacterium* (*r* = 0.20) and other 41 genera within the *Firmicutes* phylum. The statistical significance was retained after the FDR corrections (Table 2). In particular, among the main contributing food groups (of which factor loading >3) to the HiαDP, higher intake of fermented legumes was positively correlated with *Eubacterium* (*r* = 0.21, *p*-value_adj_ = 0.037) (Appendix A). The dietary patterns identified based on two other α-diversity indices (Chao1 and Faith PD) were also related to the higher intakes of fermented legumes, nuts/seeds, and tea, but not associated with the taxonomic composition of the gut microbiota (Appendix A).

### 3.3. Enterotypes of Gut Microbiota and Their Association with Dietary Intake

Enterotypes of the gut microbiota among Korean adults based on weighted UniFrac and Bray-Curtis are shown in Figure 3. Three different enterotypes were identified (Figure 3A,B). Each of these enterotypes was characterized by one of the following dominant genera; *Bacteroides, Prevotella*, and *Ruminococcus* (Figure 3C,D). The *Ruminococcus* enterotype showed significantly higher scores of the HiαDP and was more strongly associated with intakes of vegetables, seaweeds, and nuts/seeds at the food group level and dietary fiber at the nutrient level compared to the two other enterotypes, regardless of which distance matrix was used (Table 3). There were no significant differences with respect to sex, age, BMI, or smoking status (Appendix A). We also identified two enterotypes based on unweighted UniFrac and each enterotype was dominant by *Bacteroides* and *Prevotella* (Appendix A). None of the enterotypes was associated with intakes of food groups and macronutrients (data not shown).

## 4. Discussion

In the NAS-IARC cross-sectional study among Korean adults, we found that a traditional Korean dietary pattern characterized by higher intakes of plant-based and fermented foods and lower intakes of noodle products and carbonated and sugar-sweetened beverages were favorably associated with gut microbial composition and diversity. Specifically, this HiαDP was positively associated with some genera within the *Firmicutes* phylum, such as *Lactobacillus*, *Ruminococcus*, and *Eubacterium*. Further, we identified three distinct enterotypes, which were characterized by differences in the HiαDP and habitual intakes of specific foods and nutrients.

Our findings on dietary pattern characterized by higher intake of plant-based foods are in line with previous studies reporting that the gut microbial diversity of populations consuming plant-based diets in rural areas in Africa and South America was greater compared to western populations [12,41,42]. Plant-based diets are rich in dietary fiber, which is the main source of microbiota-accessible carbohydrates (MACs), a major energy source for the gut microbiota [43]. An animal model study showed that low MACs diets led to an irreversible depletion of gut microbiota diversity [44]. MACs can be metabolized by “fiber-degrading bacteria” such as *Roseburia*, *Lactobacillus*, *Eubacterium*, *Ruminococcus*, and *Bifidobacterium*, mostly belonging to the *Firmicutes* and *Actinobacteria* phyla [11,45]. Human intervention studies also found that intakes of MACs-rich foods such as whole-grain foods enhanced the presence of some bacteria of the *Firmicutes* phylum [46,47]. Consistent with these previous studies, we found that the plant-based dietary pattern in Korean adults was positively associated with some genera within the *Firmicutes* phylum including *Lactobacillus*, *Ruminococcus*, and *Eubacterium*, known as “fiber-degrading bacteria”. These fiber-degrading bacteria produce short-chain fatty acids including butyrate in the human intestine [48], which are inversely associated with obesity, diabetes, and colorectal cancer [49,50].

Another important finding in the current study was the association of fermented foods such as fermented legumes and seaweeds with taxonomic composition and diversity of the gut microbiota, which has been rarely investigated in previous studies. Fermented foods such as fermented vegetables (e.g., Kimchi), and fermented legume products, mainly based on soybean, (e.g., Cheonggukjang, Doenjang, and Ganjang) are typical dishes of traditional Korean diets [17]. These fermented foods contain living microorganisms including probiotic bacteria (e.g., *Lactobacillus*, *Bifidobacterium*, or *Streptococcus*) and bioactive compounds generated during the fermentation process which could affect the microbial composition and diversity in the human gut [19]. Seaweeds are also typical ingredients of traditional Korean dishes, including laver (Gim in Korea), sea mustard (Miyeok in Korean), and kelp (Dasima in Korean), and a source of bioactive compounds including polysaccharides, dietary fiber, polyphenols, and PUFAs [51]. Especially, seaweed polysaccharides (e.g., alginate, laminarin, and fucoidan) could regulate gut microbiota, stimulating SCFA-producing bacteria such as *Lactobacillus*, *Streptococcus*, and *Bifidobacteria*, which might have potential health benefits against metabolic diseases and certain cancers [52,53,54,55]. However, to date, there is very limited evidence on the impact of habitual intake of fermented foods and seaweeds on the gut microbiota. To the best of our knowledge, this is the first observational study investigating associations between fermented legumes and seaweeds as part of habitual diets and gut microbial composition and diversity. Besides, in our study population, the group of fermented legumes, mainly fermented soybean pastes, was one of the major components of the HiαDP, which was positively associated with the *Firmicutes* phylum and its genus *Lactobacillus*, *Ruminococcus*, and *Eubacterium*, while this association was not found in the non-fermented legume intake. Therefore, this difference may come from the fermentation process. In previous studies on microbial communities in Korean fermented soybean pastes [56,57], the dominant microbes were found to be *Bacillus* and other lactic acid bacteria, which mostly belong to the *Firmicutes* phylum. An intervention study [58] which examined the effect of a typical Korean diet including fermented foods, like Kimchi, and American-style diets on the gut microbiota in 61 overweight/obese Korean adults, also showed that the *F/B* ratio and some genera within the *Firmicutes* phylum including *Weissella* increased after consumption of the typical Korean diet, explained as the effect of the fermented food intake. However, in our study no significant association was found between higher intakes of fermented vegetables including Kimchi and the gut microbiota. This indicated that fermented vegetables like Kimchi, which is one of the most frequently (almost every day and with every meal) and widely consumed foods in the population, could not be a discriminant dietary factor in gut microbial diversity and composition among the Korean population.

In addition, we explored gut microbial enterotypes in Korean adults applying a modified multivariate cluster analysis [8]. The retained enterotypes and their dominant genera, *Bacteroides*, *Prevotella*, and *Ruminococcus,* respectively, were similar to the previous study [8]. Another study in American adults [21] suggested that gut microbial enterotypes were strongly associated with long-term diets compared to short-term diets, showing that the *Bacteroides* enterotype was strongly associated with a high intake of protein and animal fat and the *Prevotella* enterotype with a high intake of carbohydrate. Our study also showed that the enterotypes were strongly associated with habitual diet, but there was no significant difference in the intakes of protein, animal fat, and carbohydrate across enterotypes. In our study, there were significant differences in the intake of plant-based foods high in dietary fiber like vegetables, seaweeds, and nuts/seeds across enterotypes. In particular, subjects of the *Ruminococcus* enterotype were more adherent to the HiαDP, which was characterized by higher intakes of vegetables, seaweeds, nut/seeds, and dietary fiber compared to the two other enterotypes.

A major strength of our study was that it provides a first comprehensive overview of the Korean diet associated with the composition and diversity of the human gut microbiota, considering not only individual intakes of food groups and nutrients but also specific dietary patterns. The dietary pattern analysis accounted for synergistic and correlated effects of food groups [59] on the host diet-gut microbiota. In this study, the within- and between-sample diversity of the gut microbiota was estimated using different diversity indices. Based on our results, the Shannon index, i.e., the α-diversity index estimating both species richness and evenness [31], and the weighted Unifrac and Bray-Curtis distance matrix, i.e., the β-diversity indices considering the relative abundances of species [34,35], seem to reflect the effect of diet on gut microbiota better than other indices. However, a future methodological study is needed. The relatively large sample size is another strength compared to previous microbiome studies (mostly less than 100). 

The findings of this study should be interpreted in the light of the following limitations. The concept of enterotypes is controversial. A recent opinion paper [60] stated that analytical factors such as types of beta-diversity distance matrices or criteria such as the optimal number of clusters could affect the enterotype stratification, which may hamper reproducibility. In our study, therefore, we used three different distance matrices and three different optimal number criteria, and we obtained consistent enterotypes within our study population based on two different distance matrices (weighted UniFrac and Bray-Curtis). We believe that our study with a relatively large sample size (*n* = 222) can provide additional insights to improve the enterotype concept. The 16s rRNA sequencing may be associated with measurement error including limited resolution and lower sensitivity compared to metagenomic sequencing data, even though it enables the capture of broad snapshots to understand the gut microbial community in the human gut [61]. While we adjusted for confounding by factors known to affect the gut microbiota such as sex, age, BMI, intake of dietary supplements including probiotics, physical activity, and smoking status, we were unable to account for the mode of birth delivery and other unmeasured potential confounders due to lack of data [62]. Last, since this was a cross-sectional study with convenience sampling from a southern part of Korea, we cannot determine a causal relationship, and generalization of the study findings should be made cautiously.

## 5. Conclusions

We conclude that a habitual Korean diet characterized by higher intakes of fermented legumes, vegetables, potatoes, seaweeds, nuts/seeds, and tea, and lower intakes of noodle products and carbonated and sugar-sweetened beverages, was associated with a higher α-diversity and several of the genera within *Firmicutes* including *Lactobacillus*, *Ruminococcus*, and *Eubacterium*. Of three identified distinct β-diversity enterotypes, labelled according to their dominant genera, the *Ruminococcus* enterotype was associated with higher intakes of vegetables, seaweeds, legumes, nuts/seeds, and dietary fiber compared to the *Prevotella* or *Bacteroides* enterotypes. Further studies are needed to confirm our findings and investigate potential health effects of observed diet-gut microbiota associations.

## Figures and Tables

**Figure 1 nutrients-13-00366-f001:**
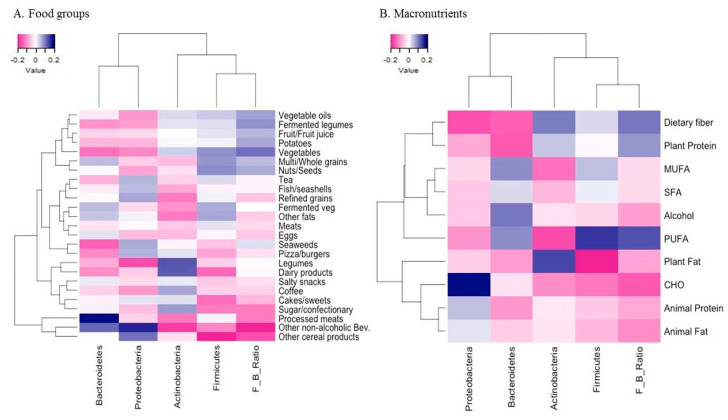
Heatmaps of partial Spearman correlations between the relative abundance of four major phyla and the *Firmicutes-to-Bacteroidetes* (*F/B*) ratio of the gut microbiota and the intakes of food groups (**A**) and macronutrients (**B**) in Korean adults of the NAS-IARC cross-sectional study (*n* = 222 participants). Partial Spearman correlation analysis adjusted for sex, age, BMI, dietary supplement intake, physical activity, smoking status, and sample batch; Intakes of food groups and macronutrients were log-transformed and adjusted for total energy intake using the residual method; the relative abundance of four major phyla and the *F/B* ratio of the gut microbiota were centered log-ratio transformed.

**Figure 2 nutrients-13-00366-f002:**
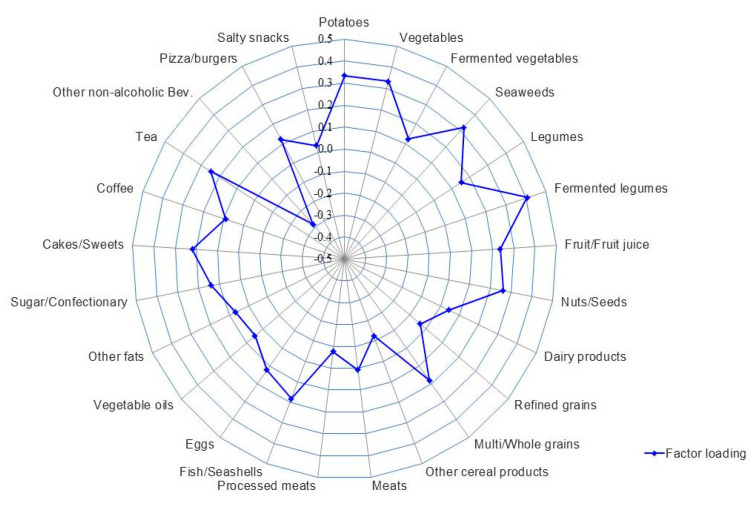
The high α-diversity dietary pattern (HiαDP) in Korean adults of the NAS-IARC cross-sectional study (*n* = 222 participants). The factor loading of each food group of the HiαDP in the Korean adults was estimated in a Reduced Rank Regression (RRR) model with the intake of 25 food groups as predictor variables and the Shannon index (α-diversity index) as a response variable; the Shannon index was log-transformed and the intakes of food groups were log-transformed and adjusted for total energy intake using the residual method.

**Figure 3 nutrients-13-00366-f003:**
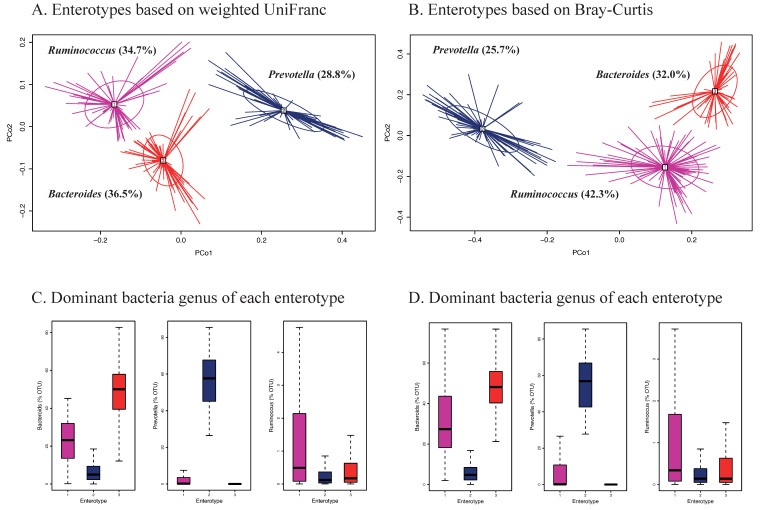
Enterotypes of gut microbiota in Korean adults of the NAS-IARC cross-sectional study (*n* = 222 participants). Three enterotypes were identified by Principal Coordinates Analysis (PCoA) and k-means clustering based on (**A**) weighted UniFrac and (**B**) Bray-Curtis distance matrix (β-diversity) of gut microbiota in the Korean adults; dominant bacteria genus of each enterotype—*Bacteroides*, *Prevotella*, and *Ruminococcus* based on (**C**) weighted UniFrac and (**D**) Bray-Curtis distance matrix; panels A and B show scores of PCo1 (Principal Coordinate) and PCo2 on the axes and each study participant’s ID is represented in the panels (legibility of ID numbers not required).

**Table 1 nutrients-13-00366-t001:** General characteristics of Korean adults of the National Institute of Agricultural Sciences of Korea and the International Agency for Research on Cancer (NAS-IARC) cross-sectional study (*n* = 222 participants).

	Total (*n* = 222)		Men (*n* = 108)		Women (*n* = 114)	
Age ^1^ (years)	29.6	20–51	26.9	21–48	32.2	20–51
BMI ^1^ (kg/m^2^)	22.9	19.1–28.5	23.6	20.2–28.8	22.3	18.8–27.0
Alcohol intake ^1^ (g/day)	9.7	0.0–39.8	14.1	0.0–52.9	5.5	0.0–27.1
Dietary supplement intake (*n*, %)						
Yes	76	34.2%	26	24.1%	50	43.9%
No	143	64.4%	80	74.1%	63	55.3%
Don’t know	3	1.4%	2	1.9%	1	0.9%
Regular physical activity (*n*, %)						
Yes	92	41.4%	49	45.4%	43	37.7%
No	130	58.6%	59	54.6%	71	62.3%
Smoking status (*n*, %)						
Ever (former/current)	54	24.3%	47	43.5%	7	6.1%
Never	168	75.7%	61	56.5%	107	93.9%
Education (*n*, %)						
<University graduation	110	49.5%	60	55.6%	50	43.9%
≥University graduation	112	50.5%	48	44.4%	64	56.1%
Household Income (*n*, %)						
<4000 USD/month	79	35.6%	40	37.0%	39	34.2%
≥4000 USD/month	93	41.9%	41	38.0%	52	45.6%
Don’t know	50	22.5%	27	25.0%	23	20.2%

^1^ Mean and range (5–95 percentiles).

**Table 2 nutrients-13-00366-t002:** Spearman correlations ^1^ between the high α-diversity dietary pattern (HiαDP) ^2^ and relative abundance (% operational taxonomic units (OTU)) of gut microbiota at the phylum and genus levels in Korean adults of the NAS-IARC cross-sectional study (*n* = 222 participants).

Phylum			Genus			
Taxa ^3^	Coefficient	*p*-Value	Taxa ^3^	Coefficient	*p*-Value ^4^	*p*-Value_adj_^4^
*F/B* ratio	0.237	0.0004				
*Bacteroidetes*	−0.170	0.012	*Coenonia*	−0.271	0.0001	0.003
			*Prevotella*	−0.222	0.001	0.006
			*Tannerella*	−0.215	0.002	0.007
*Firmicutes* ^5^	0.079	0.249	*Lactobacillus*	0.228	0.0007	0.005
			*Ruminococcus*	0.214	0.002	0.007
			*Eubacterium*	0.202	0.003	0.008
			*Hydrogenoanaerobacterium*	−0.268	0.0001	0.003
			*Desulfotomaculum*	−0.257	0.0001	0.003
			*Alkalibaculum*	−0.249	0.0002	0.003
			*Peptoniphilus*	−0.248	0.0002	0.003
			*Lactonifactor*	−0.248	0.0002	0.003
			*Acetivibrio*	−0.247	0.0003	0.003
			*Peptostreptococcus*	−0.242	0.0003	0.004

^1^ Partial Spearman correlation analysis adjusted for sex, age, BMI, dietary supplement intake, physical activity, smoking status, and sample batch; ^2^ The high α-diversity dietary pattern (HiαDP) based on the Shannon index from Reduced Rank Regression (RRR) analysis was characterized by high intakes of fermented legumes, vegetables, potatoes, seaweeds, nuts/seeds, and tea and low intake of non-alcoholic beverages (e.g., carbonated and sweet beverages); ^3^ The relative abundance of taxa and the *Firmicutes-to-Bacteroidetes* (*F/B*) ratio were centered log-ratio transformed after imputing zeros in the dataset based on a Bayesian-multiplicative replacement; ^4^
*P*-values before (*p*-value) and after (*p*-value_adj_) multiple comparison corrections using False Discovery Rate (FDR); ^5^ Other 34 significant genera within the *Firmicutes* phylum [|coefficients(r)| < 0.24] are presented in Appendix A.

**Table 3 nutrients-13-00366-t003:** Difference ^1^ in the high α-diversity dietary pattern (HiαDP) and intakes of food groups and nutrients among three enterotypes of Korean adults of the NAS-IARC cross-sectional study (*n* = 222 participants).

	*Prevotella*		*Bacteroides*		*Ruminoc* *occus*		*p*-Value
	Mean	IQR	Mean	IQR	Mean	IQR	
Weighted UniFrac	(*n* = 64, 28.8%)		(*n* = 81, 36.5%)		(*n* = 77, 34.7%)		
Dietary pattern							
HiαDP score	−0.27	−0.82–0.40	0.02	−0.40–0.62	0.20	−0.34–0.90	0.008
Food groups ^2^							
Potatoes	28.9	7.6–35.3	31.6	11.8–38.4	33.8	12.7–40.9	0.344
Vegetables	125.5	57.3–162.8	159.1	74.3–201.9	177.8	66.2–221.0	0.039
Fermented vegetables	86.6	33.7–151.9	81.6	31.2–110.8	99.7	29.1–154.0	0.250
Seaweeds	1.0	0.4–1.3	1.5	0.6–2.1	1.7	0.4–1.8	0.003
Legumes	33.6	12.1–39.0	51.3	17.2–65.1	46.8	16.2–57.2	0.030
Fermented legumes	3.2	0.7–3.7	3.8	1.3–5.1	4.2	1.3–5.1	0.160
Fruit/Fruit juice	214.5	45.1–248.8	194.7	52.1–250.6	217.3	65.4–196.9	0.525
Nuts/Seeds	2.0	0.0–2.7	2.9	0.0–4.8	4.1	0.1–5.9	0.040
Dairy products	121.6	49.5–178.2	158.0	50.0–193.1	140.1	53.2–178.8	0.645
Refined grains	437.8	328.0–641.4	450.1	321.9–651.8	440.4	429.1–440.1	0.951
Multi/Whole grains	3.6	0.0–6.0	3.8	0.0–6.0	5.2	0.0–7.2	0.238
Other cereal products	81.7	43.4–98.5	78.4	44.5–103.0	71.6	38.4–100.4	0.665
Meats	106.3	53.9–121.8	109.7	48.3–129.3	104.9	50.9–121.0	0.591
Processed meats	7.3	1.5–8.6	8.4	1.7–10.0	5.8	0.7–8.6	0.060
Fish/Seashells	31.1	14.2–38.1	37.9	17.0–49.9	36.5	17.0–44.6	0.430
Eggs	22.5	9.4–31.1	25.4	9.2–41.7	24.4	11.4–40.3	0.928
Vegetable oils	1.8	0.8–2.0	2.0	1.0–2.5	1.9	1.1–2.3	0.114
Other fats	0.5	0.1–0.6	0.7	0.1–0.8	0.5	0.1–0.4	0.460
Sugar/Confectionary	4.0	1.1–4.3	3.5	1.2–5.1	3.3	1.2–3.8	0.786
Cakes/Sweets	16.1	3.5–13.4	18.3	4.3–22.3	16.1	4.6–21.2	0.247
Coffee	4.0	0.2–5.4	2.4	0.2–2.7	2.7	0.2–5.4	0.432
Tea	22.7	0.0–10.0	32.8	0.0–20.0	17.9	0.0–12.9	0.557
Other non-alcoholic Bev.	120.5	19.2–115.2	97.6	20.8–117.8	55.8	16.7–55.3	0.238
Pizza/burgers	16.8	6.7–16.7	19.2	6.7–25.0	17.6	6.7–25.0	0.511
Salty snacks	7.5	0.0–6.4	8.1	1.0–12.9	7.4	1.2–6.4	0.467
Macronutrients ^2^							
Plant protein	34.7	24.5–39.2	36.1	26.6–41.6	36.3	27.1–43.0	0.083
Animal protein	34.6	20.2–39.2	38.7	20.2–46.5	36.3	23.3–41.7	0.287
Carbohydrate	313.0	222.0–333.1	312.5	231.0–373.2	301.7	224.7–349.7	0.903
Dietary fiber	16.4	9.5–20.1	17.5	10.5–22.3	18.5	10.7–21.9	0.037
Plant fat	17.4	9.0–21.5	17.9	10.6–21.0	17.3	11.0–22.3	0.343
Animal fat	30.4	15.6–32.8	30.5	15.4–37.4	28.8	17.2–33.5	0.619
SFA	12.1	6.4–14.0	11.4	6.6–13.1	11.6	6.9–14.3	0.365
MUFA	12.5	6.6–14.0	11.6	6.2–13.4	12.0	7.0–14.9	0.290
PUFA	5.4	3.1–5.8	5.5	3.1–6.4	5.7	3.5–7.2	0.059
Alcohol	11.1	1.1–11.5	9.5	0.5–8.3	8.6	0.7–12.7	0.465
Bray-Curtis	(*n* = 71, 25.7%)		(*n* = 57, 32.0%)		(*n* = 94, 42.3%)		
Dietary pattern							
HiαDP score	−0.23	−0.80–0.37	−0.15	−0.87–0.45	0.26	−0.31–0.94	0.005
Food groups ^2^							
Potatoes	27.7	8.9–30.1	28.2	11.8–41.7	36.6	12.7–40.9	0.148
Vegetables	123.4	58.1–163.7	132.3	57.4–151.9	194.7	74.3–242.0	0.004
Fermented vegetables	82.6	29.6–130.1	78.5	27.6–104.2	101.0	34.4–139.3	0.485
Seaweeds	1.0	0.4–1.3	1.3	0.4–1.6	1.8	0.6–2.6	0.000
Legumes	32.6	12.5–39.0	37.2	13.4–52.6	58.3	18.8–74.8	0.001
Fermented legumes	3.1	0.7–4.0	3.2	0.7–3.4	4.6	1.3–5.1	0.007
Fruit/Fruit juice	204.9	51.9–237.7	175.1	55.2–202.5	230.9	68.3–255.3	0.241
Nuts/Seeds	2.0	0.0–3.5	2.4	0.0–3.3	4.2	0.1–6.0	0.041
Dairy products	122.5	45.6–178.8	135.4	48.0–178.8	159.0	52.1–187.4	0.938
Refined grains	418.3	220.1–440.0	470.2	331.1–643.8	445.6	426.1–643.6	0.319
Multi/Whole grains	3.6	0.0–5.7	3.9	0.0–5.3	4.9	0.0–7.2	0.347
Other cereal products	79.4	43.6–95.3	85.3	50.5–112.3	70.2	34.4–96.2	0.172
Meats	103.0	51.5–125.9	99.7	47.3–123.3	114.6	55.3–126.5	0.642
Processed meats	7.2	1.3–8.6	8.5	1.7–8.6	6.4	0.7–8.6	0.087
Fish/Seashells	29.9	12.6–37.8	31.1	14.1–36.5	42.4	18.3–54.2	0.035
Eggs	22.5	11.0–30.8	22.5	7.1–29.7	26.6	9.9–41.9	0.667
Vegetable oils	1.7	0.8–2.0	1.9	1.2–2.5	2.1	1.0–2.5	0.101
Other fats	0.5	0.1–0.6	0.6	0.1–0.7	0.6	0.1–0.7	0.815
Sugar/Confectionary	4.2	1.1–6.0	3.3	1.1–4.0	3.2	1.2–4.4	0.347
Cakes/Sweets	15.7	3.7–13.8	19.3	4.3–25.6	16.3	4.3–20.9	0.579
Coffee	3.7	0.2–5.4	2.1	0.1–2.7	2.9	0.6–5.4	0.962
Tea	20.5	0.0–12.9	14.7	0.0–10.0	34.0	0.0–25.7	0.188
Other non-alcoholic Bev.	112.4	20.8–112.5	96.3	21.7–110.7	68.6	16.7–75.0	0.053
Pizza/burgers	16.2	6.7–16.7	20.9	6.7–25.0	17.6	6.7–25.0	0.714
Salty snacks	7.2	0.0–6.4	10.8	0.5–12.9	6.2	1.0–7.5	0.733
Macronutrients ^2^							
Plant protein	33.4	22.9–37.2	35.5	25.6–39.2	37.7	28.6–44.4	0.001
Animal protein	33.7	19.7–38.0	34.6	21.0–40.0	40.2	22.7–45.6	0.208
Carbohydrate	301.3	210.1–322.7	314.1	234.1–351.7	311.4	235.3–365.2	0.401
Dietary fiber	15.9	9.3–19.7	16.3	9.8–19.9	19.5	11.9–23.7	0.002
Plant fat	16.7	8.6–21.2	18.3	11.1–20.7	17.7	10.9–23.8	0.202
Animal fat	29.6	15.3–33.0	28.0	15.9–35.2	31.2	17.3–37.4	0.848
Saturated fatty acids	11.8	6.3–13.9	10.5	6.9–13.1	12.3	6.6–14.8	0.516
Monounsaturated fatty acids	12.2	6.3–13.9	10.5	6.2–13.4	12.7	6.7–15.0	0.401
Polyunsaturated fatty acids	5.2	3.1–5.7	5.0	3.2–6.2	6.1	3.5–7.6	0.011
Alcohol	10.3	0.5–11.0	8.1	0.7–9.4	10.1	0.7–9.8	0.591

HiαDP, high α-diversity dietary pattern; IQR, interquartile range; ^1^ Difference in dietary pattern scores and intakes of food groups and nutrients among three enterotypes were examined by general linear models (GLMs) with sex, age, BMI, dietary supplement intake, physical activity, smoking status, and sample batch as covariates; ^2^ The intakes of food groups and nutrients were log-transformed and adjusted for total energy intake using the residual method.

## Data Availability

The 16S rRNA gene sequencing data supporting the conclusion of this study are available in NCI Sequence Read Archive (SRA) with study accession number, BioProject ID: PRJNA644479.

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
