# Peer review of "Taxonomic Composition and Diversity of the Gut Microbiota in Relation to Habitual Dietary Intake in Korean Adults"

_nutrients, 2021, doi:10.3390/nu13020366_

Round 1

Reviewer 1 Report

Overall the study design including sample collection was appropriate. Both bioinformatic and statistical analyses were comprehensive and clearly described. 

However, the study had some major weaknesses including:

  1. Most analyses were based on the design of classifying  gut microbiota into three distinct “enterotypes”. The concept of enterotypes is controversy. The analysis based on it could be arguable.
  2. F/B is old school measurement. Phylum level is higher, the results based on F/B may be not clinically relevant. 
  3. The correlation analysis of relative data could result in spurious correlation. 

Minor:

There are few typos in the text and references, which should be corrected before publishing.

Author Response

Response to Reviewer 1 Comments

We appreciate the reviewers for their comments. We have revised the manuscript according to the comments by highlighting them using the “Track Changes” function in the revised version. Our point-by-point responses to the comments are given below.

Major Comment 1: Most analyses were based on the design of classifying gut microbiota into three distinct “enterotypes”. The concept of enterotypes is controversy. The analysis based on it could be arguable.

Response: We thank the Reviewer for this comment. We acknowledge that the concept of enterotypes is controversial. Consequently, we briefly discuss the current limitations of the enterotype concept in the Discussion section (Page 13, lines 427-434). We also added an opinion paper (PMID: 31026581), which reviewed the concept of enterotypes.

Major Comment 2: F/B is old school measurement. Phylum level is higher, the results based on F/B maybe not clinically relevant. 

Response: We thank for this comment and also acknowledge the Reviewer’s point. In previous studies, the F/B ratio has been used as a taxonomic signature for obesity and other related conditions. However, as recently addressed in a review paper (PMID: 32438689), the F/B ratio as a clinical indicator might be limited since many lifestyle factors (e.g. diet, physical activity, and antibiotics) could affect the ratio. Our study also demonstrated that a specific dietary pattern in the study population was associated with the F/B ratio. Besides, as the Reviewer pointed out, the Phylum is the highest taxonomic level of the gut microbiota, so it’s useful to see a global trend but less informative than other lower levels. Therefore, we emphasized the genus level rather than the phylum level or the F/B ratio throughout the revised manuscript.     

Major Comment 3: The correlation analysis of relative data could result in spurious correlation. 

Response: We thank the Reviewer for this comment. In full agreement, we implemented additional analyses for the compositional data (PMID: 31544212). We imputed zeros in the compositional count data based on a Bayesian-multiplicative replacement and then used centered log-ratio transformation. We revised the manuscript according to the new statistical analysis in the Method (Pages 4-5, lines 176-225) and Result (Pages 6-9, Sections 3.1 and 3.2) sections including Figure 1, Table 2, and supplemental tables (Tables S5-S7).     

Minor Comment 1: There are few typos in the text and references, which should be corrected before publishing. 

Response: We checked and revised all typos in the text and references.

Reviewer 2 Report

The manuscript entitled “Taxonomic composition and diversity of the gut  microbiota in relation to habitual dietary intake in Korean adults’ prepared by Noh et al. presents interesting information based on correlation between gut microbiota and habitual dietary intake. Basis of the studies were 222 Koreans aged 18-58 years. Obtained results revealed significant relation between aforementioned factors. On the whole, manuscript is well prepared. Scientists used adequate methods, studies were well planned as well as statistical analysis, which play a pivotal role, was performed in detail. References are well selected. This manuscript fits the scope of the field covered by the journal "Nutrition", the following point has to be improved:

  1. Figures (especially Figure 3.) are illegible. Please correct font, size and form of presentation of obtained results because in present form there is too much detail results which are unreadable.

Author Response

Response to Reviewer 2 Comments

We appreciate the reviewers for their comments. We have revised the manuscript according to the comments by highlighting them using the “Track Changes” function in the revised version. Our point-by-point responses to the comments are given below.

Comment 1: Figures (especially Figure 3.) are illegible. Please correct font, size and form of presentation of obtained results because in present form there is too much detail results which are unreadable.

Response: We thank the reviewer for the comment. We revised the figures and increased the font size of the text, and submitted separate files in a different figure format (.jpg) for editing.

Reviewer 3 Report

The manuscript presents a relevant cross-sectional study that provides a innovative, robust and comprehensive overview of the Korean diet associated with the composition and diversity of the human gut microbiota, considering not only individual intakes of food groups and nutrients but also specific dietary patterns.

Only Minor revisons should be taken into account before its publication:

Please revise and correct Line 133 subheading: 2.4.16. s rRNA gene sequencing and taxonomic assignment

Please revise and correct OUT by OTU of "Table 2. Spearman correlations1 between the high α-diversity dietary pattern (HiαDP)2 and 285 relative abundance (% OUT)"..

Please draw a conclusion less assertive and show the main microbial associations findings.

Author Response

Response to Reviewer 3 Comments

We appreciate the reviewers for their comments. We have revised the manuscript according to the comments by highlighting them using the “Track Changes” function in the revised version. Our point-by-point responses to the comments are given below.

Comment 1 & 2: Please revise and correct Line 133 subheading: 2.4.16. s rRNA gene sequencing and taxonomic assignment

Please revise and correct OUT by OTU of "Table 2. Spearman correlations1 between the high α-diversity dietary pattern (HiαDP)2 and 285 relative abundance (% OUT)"

Response: We thank the Reviewer for spotting these two typos above. They have been corrected accordingly.

Comment 3: Please draw a conclusion less assertive and show the main microbial associations findings.

Response: We thank the Reviewer for this comment. We have revised the conclusions in the abstract (Page 1, lines 37-39) and discussion (Page 13, lines 444-455) sections accordingly reflecting our main findings.

Round 2

Reviewer 1 Report

All my comments have been either addressed or discussed as limitations. 

Author Response

We appreciate the reviewer for accepting our revisions